# Development and Validation of the Meiji Nutritional Profiling System per Serving Size

**DOI:** 10.3390/nu16162700

**Published:** 2024-08-14

**Authors:** Ryota Wakayama, Adam Drewnowski, Tomohito Horimoto, Tao Yu, Yoshie Saito, Takao Suzuki, Keiko Honda, Shigehiko Kanaya, Satoshi Takasugi

**Affiliations:** 1Meiji Co., Ltd., 2-2-1 Kyobashi, Chuo-ku, Tokyo 104-9306, Japan; tomohito.horimoto@meiji.com (T.H.); tao.yu@meiji.com (T.Y.); satoshi.takasugi@meiji.com (S.T.); 2Computational Systems Biology Laboratory, Division of Information Science, Graduate School of Science and Technology & Data Science Center, Nara Institute of Science and Technology, 8916-5 Takayama-cho, Ikoma 630-0912, Japan; 3Center for Public Health Nutrition, University of Washington, Seattle, WA 98195, USA; 4Meiji Holdings Co., Ltd., 2-4-16, Kyobashi, Chuo-ku, Tokyo 104-0031, Japan; 5National Center for Geriatrics and Gerontology, 7-430 Morioka, Obu 474-8511, Japan; 6Laboratory of Medicine Nutrition, Kagawa Nutrition University, 3-9-21 Chiyoda, Sakado 350-0288, Japan

**Keywords:** nutrient profiling, nutrient-rich foods index, convergent validity, hybrid nutrient density score, serving size, nutritional intake

## Abstract

Serving size may be the appropriate reference for calculating food nutritional value. We aimed to assess the nutritional values of Japanese foods based on serving sizes rather than per 100 g by adapting the Meiji Nutritional Profiling System (Meiji NPS). Given the variability in serving sizes across countries, we used Japanese serving sizes to calculate the Meiji NPS scores. We confirmed the convergent validity of the Meiji NPS scores per serving size with the Nutrient-Rich Food Index 9.3 using Spearman’s correlation coefficients (r = 0.51, *p* < 0.001). Food groups recommended by official guidelines, such as pulses, nuts and seeds, fish and seafood, fruits, vegetables, and milk and milk products, scored relatively high. Furthermore, the nutrient density scores of food items with small serving sizes, such as mushrooms, algae, seasonings, and fats and oils, were moderated when calculated by per serving size, despite having considerably higher or lower scores per 100 g. These results indicate that calculating NPS per serving size allows for the assessment of the nutritional value of food items in accordance with actual consumption quantities. Therefore, the Meiji NPS calculated per serving size, alongside the per 100 g version, may be useful for dietary management depending on specific purposes.

## 1. Introduction

Nutrient profiling (NP) comprises a set of quantitative methods developed to assess the nutrient density of individual foods, dishes, or meals [1,2]. The nutrient density of individual foods is typically calculated as the content of multiple nutrients per reference amount, which can be 100 g, 100 kcal, or a serving size [2,3]. Foods are then classified, ranked, or rated based on their overall nutrient density scores.

A commonly used reference amount is 100 g. The original FSA-Ofcom NP model and its derivatives, including the Health Star Rating (HSR) and Nutri-Score NP systems, use 100 g as the basis for calculating the nutrient density of foods [4,5,6,7,8]. One reason for this is influenced by the absence of government-mandated serving sizes in the United Kingdom and the European Union. By contrast, Health Canada provides guidance on appropriate serving sizes to ensure consistency across similar products. The front-of-package nutrition symbol in Canada is based on serving size [9,10,11,12,13]. Although the United States (US) Food and Drug Administration lists mandatory serving sizes, the US-based Nutrient-Rich Food Index (NRF) uses 100 kcal as the main basis of calculation [14,15,16] but has also been calculated per 100 g.

NP aims to compare the nutrient densities of foods across and within food groups. The World Health Organization views NP methods as useful tools in dietary interventions for health promotion and disease prevention [1]. However, NP methods designed for individual foods do not consider the differences in the actual amounts of food consumed by individuals. Not all foods are consumed in 100 g amounts. For instance, serving sizes for seasonings, fats, and oils are typically around 15 g or less, whereas those for beverages are approximately 240 g [16]. An NP model calculated per 100 g would penalize foods consumed in small amounts and favor foods consumed in large amounts; therefore, corrective algorithms are required [17].

Calculating nutritional values per serving size can be more intuitive and better reflect the consumer experience. Presenting nutritional information by serving size may make it easier for consumers to understand, enabling them to make informed decisions about their diets and lifestyles [18,19]. According to several reports, consumers prefer nutritional information per serving size as a reference amount [20].

The Meiji Nutritional Profiling System (Meiji NPS) is an age-sensitive nutritional profiling method tailored to specific health issues in adults and older adults [21,22]. Two versions of the Meiji NPS were developed: one for adults (<65 years old) and one for older adults (≥65 years old). The Meiji NPS uses 100 g of food as the basis for calculation. Recognizing the importance of serving size, the present version of the Meiji NPS calculates nutritional values per serving.

To achieve this, we first need to determine a set of Japanese serving sizes that reflect the amounts customarily consumed at any given time. Unlike in the US [20], food nutritional labels in Japan are approved to use 100 g, 100 mL, portion size, or serving size, which can vary across different food products and companies [23,24,25]. While Japan does not have government-approved serving sizes, ordinary serving values capture the average quantity of a particular food item typically consumed in a single serving [26,27]. In general, customary serving sizes in Japan tend to be smaller than those in other countries [28,29,30].

To the best of our knowledge, there are no official values for serving size in Japan, and therefore no previous study has specifically assessed the nutritional values of individual food items per serving size by the NPS in Japan. In this study, we aimed to identify Japanese serving sizes from the available literature and adapt the Meiji NPS algorithm to evaluate nutritional values per serving size in Japan and to investigate its validity.

## 2. Materials and Methods

### 2.1. Nutrient Composition Database

Nutrient composition data were obtained from the Japanese Food Standard Composition Table 2020 (8th Edition), published by the Ministry of Education, Culture, Sports, Science, and Technology, Japan [31]. This database lists 2478 food items, with energy and macro- and micronutrient values expressed per 100 g. Prepared meals were excluded, resulting in 2428 food items. Many food items had missing data, particularly for total sugar content. Following published methods [21], foods with missing nutrient values were removed, yielding an analytical database of 1545 food items. The Japanese Food Standard Composition Table does not provide data on the content of each food group to encourage (e.g., amounts of dairy or fruits). Therefore, we set the food groups to encourage 100% representation by food category, based on previous methods [21]. For food items with specified proportions of fruit, we adhered to the indicated ratios.

### 2.2. Meiji NPS for Adults and Older Adults

In the present study, we focused on Japanese adults, including those of older ages. The algorithm for the Meiji NPS per serving size used the 100 g reference amount, also measured in grams. The detailed methodology of the Meiji NPS has been previously published [21]. Briefly, the Meiji NPS for adults and older adults was based on the ratio of “nutrients to encourage”, “nutrients to limit”, and “food groups to encourage” to the age-appropriate reference daily values (Equations (1) and (2)). In the Meiji NPS for adults, protein, dietary fiber, calcium, iron, and vitamin D were the nutrients to encourage, while energy, saturated fatty acids, sugars (the sum of glucose, galactose, fructose, maltose, sucrose, and lactose), and salt equivalents were the nutrients to limit. In the Meiji NPS for older adults, protein, dietary fiber, calcium, and vitamin D were the nutrients to encourage, while energy, sugars, and salt equivalents were the nutrients to limit. For both adults and older adults, the food groups to encourage included fruits, vegetables, nuts, legumes, and dairy. This calculation incorporates the caps set by the gaps between the Dietary Reference Intakes for Japanese (2020) [32] and the National Health and Nutrition Survey [33] (Table 1). Regarding the food groups, we followed the Japanese Food Standard Composition Table 2020 (8th edition) and a prior study that classified foods based on nutrient density [31,34].
(1)$$\begin{array}{ll} Meiji\;NPS\;for & adults\\ & =\sum_{i=1}^{5}\left({nutrients\;to\;encourage}_{i}/{RDV}_{i}\right)\times 100\\ & -\sum_{i=1}^{4}\left({nutrients\;to\;limit}_{i}/{RDV}_{i}\right)\times 100\\ & +\sum_{i=1}^{5}\left({food\;groups\;to\;encourage}_{i}/{RDV}_{i}\right)\times 100 \end{array}$$

Equation (1) Meiji NPS for adults.
(2)$$\begin{array}{ll} Meiji\;NPS\;for & older\;adults\\ & =\sum_{i=1}^{4}\left({nutrients\;to\;encourage}_{i}/{RDV}_{i}\right)\times 100\\ & -\sum_{i=1}^{3}\left({nutrients\;to\;encourage}_{i}/{RDV}_{i}\right)\times 100\\ & +\sum_{i=1}^{5}\left({nutrients\;to\;encourage}_{i}/{RDV}_{i}\right)\times 100 \end{array}$$

Equation (2) Meiji NPS for older adults.

### 2.3. Serving Size

According to the book “Ordinary serving values food composition tables”, serving sizes were reported as one or more edible quantities for a single food item [26]. The representative serving size for each food item was determined by calculating the median of these quantities. Of the 1545 food items, 1099 food items had an edible quantity available. The Meiji NPS score was calculated for these 1099 food items based on their edible quantities. The representative nutrient density score for each food item was determined using the median of the Meiji NPS scores. In the absence of specific data for adults and older adults, we assumed that the serving size did not vary between these groups.

### 2.4. Statistical Analysis

The data are expressed as medians (interquartile ranges). The convergent validity of the Meiji NPS per serving size for adults and older adults was assessed by comparing it with the NRF9.3. The reference daily values for NRF9.3 were calculated based on the Dietary Reference Intakes for Japanese (2020) [32]. Spearman’s correlation test was used to compare the Meiji NPS per serving size with the Meiji NPS per 100 g or NRF9.3 [35]. All statistical analyses were conducted using R software version 4.3.1 (The R Foundation for Statistical Computing, Vienna, Austria).

## 3. Results

### 3.1. Serving Size

Serving sizes varied across different food items and food groups, ranging from 0.5 g to 500.0 g (Table 2). For the entire set of food items, the median serving size was 35.0 g (interquartile range: 20.0 g to 45.0 g), which was less than 100 g. Specifically, the median serving sizes for nuts and seeds, mushrooms, algae, milk and milk products, fats and oils, and seasonings and spices were all less than 15 g. By contrast, beverages were the only food group with a median serving size exceeding 100 g. One food item in the fish and seafood category was katsuobushi (simmered, smoked, and fermented skipjack tuna), which had a serving size of 1 g. Katsuobushi is a primary ingredient in dashi, a broth that forms the foundation of many soups and sauces in Japanese cuisine. Similarly, in the beverages category, matcha (a finely ground powder made from specially grown and processed green tea leaves) had a serving size of 1.5 g.

### 3.2. Meiji NPS for Adults per Serving Size

Comprehensive nutrient and food group data were obtained for 1099 food items. Meiji NPS scores for adults were calculated per serving size, with the distribution of these scores shown in Figure 1. The Meiji NPS scores ranged from −59.9 to 114.9. In terms of central tendency, the median Meiji NPS score for adults was 10.2. Notably, none of the food items were categorized as sugars or sweeteners due to the absence of the requisite nutrient data. Consequently, calculating the median, maximum, and minimum values for this category was impossible. The results are summarized in Table 3. The distribution and central tendency of Meiji NPS scores for older adults per serving size are shown in Appendix A.

### 3.3. Convergent Validity of the Meiji NPS for Adults and Older Adults per Serving Size

To assess the convergent validity of the Meiji NPS scores, we compared them with NRF9.3 scores for the same food items. Spearman’s correlation coefficient, which indicates the relationship between the Meiji NPS for adults per serving size and NRF9.3, was 0.51 (Table 4). Within the context of the Meiji NPS for adults per serving size, the correlation coefficients for nuts and seeds (r = −0.20), vegetables (r = 0.00), mushrooms (r = 0.38), and fish and seafood (r = −0.07) (all *p* > 0.05) were relatively low and not statistically significant. Similarly, for the Meiji NPS for older adults, the correlation coefficients for nuts and seeds (r = −0.30), vegetables (r = 0.02), mushrooms (r = 0.35), and algae (r = 0.33) (all *p* > 0.05) were also relatively low and not statistically significant (Appendix A). Spearman’s correlation coefficients between the Meiji NPS per serving size and per 100 g were found to be 0.83 and 0.85, respectively.

## 4. Discussion

This study is the first attempt to evaluate the nutrient values of Japanese food items per serving size using the Meiji NPS. The primary objective was to adapt the Meiji NPS algorithm, originally calculated per 100 g, to Japanese serving sizes. Our findings show that the Meiji NPS algorithm can be utilized not only per 100 g but also per serving size for both adults and older adults (Appendix A).

Spearman’s correlation coefficients between the Meiji NPS per serving size and NRF9.3 were 0.51 for adults and 0.45 for older adults, indicating a moderate correlation level [35]. To further validate the convergent validity of the Meiji NPS per serving size, Meiji NPS scores per serving size were compared with HSR scores for the same food items. The HSR scores were calculated using the HSR calculator [36], which evaluates food items on a 10-point scale. Correspondingly, Meiji NPS scores per serving size were divided into deciles. Additional comparisons with the HSR for convergent validity also showed a moderate correlation (r = 0.51 for adults, r = 0.50 for older adults) (Appendix A). These results suggest that the Meiji NPS per serving size is valid for both adults and older adults.

The Meiji NPS per serving size awarded high scores to food groups, such as pulses, nuts and seeds, fish and seafood, fruits, vegetables, fruits, and milk and milk products, with all scores higher than the overall median score for all food items. Similarly, the Meiji NPS per 100 g also awarded high scores to these food groups, which were higher than the median score for all food items. These scores align well with the World Health Organization’s “Healthy Diet” guidelines, which promote the consumption of vegetables, fruits, nuts and seeds, and pulses [37,38]. The consumption of milk and milk products is also recommended by the “Japanese Food Guide Spinning Top” [39,40,41] and “Health Japan 21” [42]. Diets high in the recommended food groups in the Meiji NPS have been reported to prevent obesity [43,44,45,46,47,48,49,50], hypertension [51,52,53,54], dyslipidemia [51,55,56,57,58,59,60,61,62,63], and diabetes [48,49,64,65]. Such diets have also been associated with a lower risk of all-cause and cardiovascular disease mortality among the Japanese population [66,67], as well as a decreased risk of dementia and frailty among older adults [68,69,70,71,72,73]. The Meiji NPS per serving size successfully identified food items generally associated with better health outcomes. These food groups can be considered ideal due to their high nutrient quality and their potential to contribute to better health when consumed regularly.

Comparing the Meiji NPS per serving size score with the Meiji NPS per 100 g score for the same foods yielded Spearman’s correlation coefficients of 0.83 for adults and 0.85 for older adults, suggesting a strong correlation. However, the central tendency differed between the two scores. The median scores for food groups such as mushrooms and algae were higher than the median score in the Meiji NPS per 100 g for all food items. Conversely, the median scores for these food groups were lower than the median score in the Meiji NPS per serving size for all food items. This discrepancy can be explained by the smaller serving sizes typical of these food groups. Despite their high Meiji NPS scores per 100 g, mushrooms and algae are only consumed in small quantities under normal conditions. Therefore, innovations, such as new processing methods, to make these foods easier to consume could increase their intake and potentially amplify their health benefits.

The nutrient profiles, including the Meiji NPS, can yield different results when calculated per serving versus per 100 g. For example, the Meiji NPS per serving size was higher for food groups such as fats and oils, confectionery, and seasonings and spices, which typically have small serving sizes. These food items, as well as butter and seasonings, were outliers in the Meiji NPS per 100 g, with scores lower than those of other food items. Although these items still scored low on the Meiji NPS per serving size, the disparities in score values were reduced. A moderate intake of saturated fatty acids, which are abundant in fats and oils, has been associated with a low risk of intracerebral hemorrhage and ischemic stroke in Asians, including the Japanese [74,75,76]. This suggests that even for food items high in nutrients, such as fats and oils, their negative impacts can be controlled by managing the serving size.

Beverages with large serving sizes (typically 240 g in the US) can pose challenges for NP models. In the Meiji NPS, fruit juices containing more nutrients and food groups to encourage received higher scores, whereas sugar-sweetened beverages containing more nutrients to limit received lower scores when evaluated per serving size compared with per 100 g. Managing added sugar is a particular issue, as sugar-sweetened beverages are the main source of sugars [77,78,79]. Increased consumption of sweetened beverages is positively associated with body weight gain and a higher risk of type 2 diabetes [80,81,82,83,84]. Conversely, consumption of fruit juice within the fruit group is associated with higher intakes of food groups to encourage and other nutrients of interest, such as vitamin C and potassium, as well as nutrients to limit [85,86,87,88,89]. Generally, 100% juices contain free sugars but not added sugars [90]. By contrast, juice-based soft drinks and nectars (with 10% to 30% juice) typically contain added sugar [91]. Beverages that are relatively easy to consume may serve as important sources of both nutrients to encourage and nutrients to limit. In other words, the reformulation of beverages may be crucial for health issues.

The Meiji NPS is designed to target the Japanese population as a whole, rather than individuals with specific conditions. This approach is consistent with other existing NPS frameworks, which also focus on populations rather than individual-specific needs. From a product reformulation perspective, nutritional information per 100 g may be more suitable due to its comparability. However, nutritional value scores per serving size may provide more relevant information to consumers and food manufacturers, aligning better with dietary guidelines. This method helps to create a useful benchmark for product reformulation for food manufacturers. Additionally, scores per serving size account for the product’s volume, potentially making it better received by consumers. Therefore, judicious use of both the Meiji NPS per 100 g and per serving size, depending on specific purposes, is crucial.

In this study, we initially calculated the Meiji NPS score per serving size based on predetermined serving sizes for food items, as outlined in available publications. Notably, no government-approved serving sizes are available in Japan. The range of serving sizes shown in Table 2 serves as the main reference for determining the serving sizes of these items. In fact, the predictive validity study of the NPS is validated according to the intake volume of the target population.

Calculating the Meiji NPS score per serving size based on consumption patterns from dietary surveys is also feasible and may better reflect the actual quantity of food consumed by individuals. Scores based on portion sizes from surveys would better accommodate individual validations in portion sizes. Consequently, the Meiji NPS algorithm per serving size can be applied to observational epidemiological research to assess diet quality in relation to health outcomes.

However, variability in serving size can be confusing from a consumer information perspective, potentially undermining the utility of nutritional information [18] as perceived and interpreted by consumers [92]. Furthermore, from the standpoint of changing consumer communication, expressing the Meiji NPS scores on an easily understandable scale, such as the Nutri-Score or HSR, may be preferable [6]. For instance, the score can be transposed onto a scale ranging from 0 (least healthy) to 100 (most healthy), thereby enhancing interpretability (Appendix A). Evaluation of food products using the Meiji NPS per serving size is speculated to provide nutritional information that aligns with the needs of different life stages.

### Limitations

This study had several limitations. First, the analysis was specifically conducted on food items listed in the Japanese Food Standard Composition Table 2020 (8th Edition), which possessed ordinary serving values. Second, the book “Ordinary serving values food composition tables” was used as a reference for serving sizes in Japan. However, the serving sizes recorded in this book may differ from the actual quantities consumed by Japanese people in a single consumption. Third, the study assumed that the serving size remained consistent between adults and older adults. However, serving sizes may vary depending on the stage of life. When applying the Meiji NPS per serving size to other life stages, such as children (<12 years old), considering the appropriate serving sizes is necessary. Additional validation studies, such as predictive studies, would strengthen the validity of the Meiji NPS per serving size.

## 5. Conclusions

The Meiji NPS algorithm is applicable to both serving size and 100 g measurements. Depending on the context, using both the Meiji NPS per 100 g and per serving size is important. The Meiji NPS per serving size may prove useful in providing nutritional information to consumers.

## Figures and Tables

**Figure 1 nutrients-16-02700-f001:**
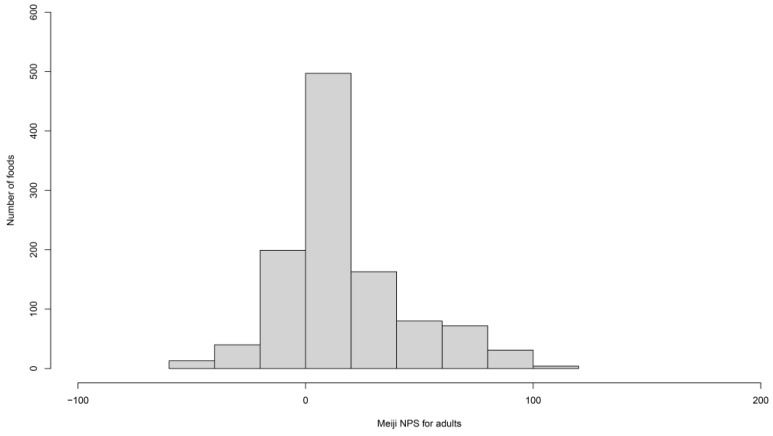
Distribution of the Meiji NPS scores for adults per serving size. The highest score was 114.9, whereas the lowest score was −59.9.

**Table 1 nutrients-16-02700-t001:** RDVs and caps of the Meiji NPS.

Items		For Adults		For Older Adults	
		RDV	Cap	RDV	Cap
Nutrients to encourage	Protein	65 g	65 g	60 g	60 g
	Dietary fiber	21 g	21 g	20 g	20 g
	Calcium	1000 mg	423.9 mg	750 mg	389.4 mg
	Iron	12 mg	5.8 mg	NA	NA
	Vitamin D	9.5 µg	6.2 µg	8.5 µg	8.5 µg
Nutrients to limit	Energy	2800 kcal	NA	2400 kcal	NA
	SFAs	31.1 g	NA	NA	NA
	Sugar	70 g	NA	60 g	NA
	Salt equivalents	7.5 g	NA	7.5 g	NA
Food groups to encourage	Fruits	200 g	200 g	200 g	113 g
	Vegetables	350 g	157.7 g	350 g	84.7 g
	Nuts	75 g	75 g	75 g	75 g
	Legumes	100 g	90 g	100 g	57 g
	Dairy	130 g	108.5 g	130 g	55 g

RDV, reference daily value; NA, not applicable; SFA, saturated fatty acid.

**Table 2 nutrients-16-02700-t002:** Summary results of serving sizes.

Items	n	Median (g)	Max (g)	Min (g)	IQR
Cereals	77	80.0	300.0	3.0	30.0 to 140.0
Potatoes and starches	18	55.0	80.0	15.0	40.0 to 60.0
Sugars and sweeteners	0	NA	NA	NA	NA
Pulses	42	35.0	175.0	4.0	15.0 to 45.0
Nuts and seeds	22	13.8	25.0	0.8	7.5 to 15.0
Vegetables	123	30.0	175.0	0.8	15.0 to 50.0
Fruits	62	50.0	175.0	5.0	26.3 to 65.0
Mushrooms	24	15.0	25.0	1.5	13.8 to 15.0
Algae	9	3.0	7.5	0.5	2.0 to 7.5
Fish and seafood	350	35.0	70.0	1.0	15.0 to 45.0
Meat	210	40.0	55.0	4.0	35.0 to 45.0
Eggs	13	22.5	50.0	10.0	10.0 to 40.0
Milk and milk products	37	15.0	175.0	6.5	12.5 to 175.0
Fats and oils	4	10.8	11.5	7.5	9.4 to 11.5
Confectionery	82	30.0	80.0	2.0	20.0 to 50.0
Beverages	10	180.0	500.0	1.5	90.0 to 180.0
Seasonings and spices	16	10.0	15.0	1.0	9.5 to 12.5
Total	1099	35.0	500.0	0.5	20.0 to 45.0

In total, 1099 food items had documented serving sizes. However, the food items categorized under sugar and sweeteners lacked complete nutrient data (indicated as “not applicable”). n, number of food items; Max, maximum; Min, minimum; IQR, interquartile range; NA, not applicable.

**Table 3 nutrients-16-02700-t003:** Results of the Meiji NPS scores for adults per serving size and per 100 g.

Items	n	Meiji NPS per Serving Size				Meiji NPS per 100 g			
		Median	Max	Min	IQR	Median	Max	Min	IQR
Pulses	42	55.7	114.9	6.6	38.0 to 69.9	161.3	278.2	66.3	116.1 to 245.1
Nuts and seeds	22	26.3	45.5	1.8	10.0 to 31.7	152.2	257.1	−9.1	134.2 to 184.2
Fish and seafood	350	25.0	102.8	−38.8	7.8 to 53.8	62.5	229.3	−155.2	27.7 to 91.1
Fruits	62	15.1	91.2	−12.4	6.5 to 28.9	39.9	77.7	−165.6	134.2 to 184.2
Vegetables	123	13.8	47.6	0.2	7.8 to 19.9	46.2	141.7	3.3	38.2 to 64.5
Milk and milk products	37	11.6	102.8	−38.8	7.8 to 53.8	54.2	83.8	−147.2	−31.7 to 74.5
Mushrooms	24	9.9	58.1	−1.1	7.8 to 13.5	66.6	275.8	−10.9	53.2 to 143.2
Potatoes and starches	18	7.5	22.8	−6.3	4.8 to 10.4	14.9	49.4	−12.6	10.9 to 20.7
Eggs	13	7.2	32.4	−1.3	2.2 to 12.4	25.9	101.3	−13.2	7.2 to 55.0
Algae	9	6.2	29.2	−2.7	4.2 to 15.0	144.2	238.9	−73.9	77.2 to 170.5
Meat	210	4.2	37.0	−53.7	−4.2 to 12.1	10.1	96.1	−119.3	−16.1 to 31.2
Cereals	77	1.8	38.7	−59.9	−0.6 to 12.4	5.5	133.1	−66.6	−0.9 to 17.2
Beverages	10	−6.8	31.9	−11.9	−11.6 to −1.5	−3.4	251.7	−6.6	−6.4 to −0.7
Seasonings and spices	16	−7.9	3.3	−27.4	−11.4 to 0.3	−72.6	219.1	−219.0	−108.1 to 3.1
Confectionery	82	−10.4	13.1	−34.3	−19.4 to −4.1	−35.5	43.5	−152.2	−60.4 to −18.9
Fats and oils	4	−13.8	0.0	−23.4	−21.6 to −5.0	−135.6	−50.2	−203.6	−187.6 to −79.3
Sugars and sweeteners	0	NA	NA	NA	NA	NA	NA	NA	NA
Total	1099	10.2	114.9	−59.9	0.7 to 25.9	36.4	278.2	−219.0	1.7 to 71.5

The Meiji NPS score per serving size for adults was calculated for 1099 food items. The food categories were based on the food groups listed in the Japanese Food Standard Composition Table 2020 (8th Edition) [31]. n, number of food items; Max, maximum; Min, minimum; IQR, interquartile range; NA, not applicable.

**Table 4 nutrients-16-02700-t004:** Spearman’s correlation coefficients between the Meiji NPS for adults per serving size and the NRF9.3 nutrient density score.

Items	n	r	*p*-Values
Cereals	77	0.83	<0.001
Potatoes and starches	18	0.80	<0.001
Sugars and sweeteners	0	NA	NA
Pulses	42	0.34	0.030
Nuts and seeds	22	−0.20	0.380
Vegetables	123	0.00	0.960
Fruits	62	0.23	0.067
Mushrooms	24	0.38	0.066
Algae	9	0.53	0.148
Fish and seafood	350	−0.07	0.185
Meat	210	0.89	<0.001
Eggs	13	0.65	0.018
Milk and milk products	37	0.65	<0.001
Fats and oils	4	0.80	0.333
Confectionery	82	0.39	<0.001
Beverages	10	0.89	0.001
Seasonings and spices	16	0.87	<0.001
Total	1099	0.51	<0.001

Sugar and sweetener data were not available. NPS, nutritional profiling system; NA, not available.

## Data Availability

The food composition data presented in this study are openly available in the Japanese Food Standard Composition Table 2020 Edition (8th Edition) at https://www.mext.go.jp/a_menu/syokuhinseibun/mext_01110.html (accessed on 28 March 2024). The ordinary serving values data presented in this study are available in the book “[Ordinary serving values food composition tables] jouyouryou shokuhin seibun hayami hyou (in Japanese)” published by Ishiyaku Publishers, Inc.

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
