# Peer review of "Development and Validation of the Meiji Nutritional Profiling System per Serving Size"

_nutrients, 2024, doi:10.3390/nu16162700_

Round 1

Reviewer 1 Report

Comments and Suggestions for Authors

The paper “Development and Validation of the Meiji Nutritional Profiling System per Serving Size” aimed to assess the nutritional values of Japanese foods based on serving sizes rather than per 100 g by adapting the Meiji Nutritional Profiling System (Meiji NPS). The research is interesting and meaningful, but there are some questions that need to be further improved or explained.

Comments:

Q1. The excessive number of keywords necessitates further simplification.

Q2. The abstract fails to accurately represent the target population and specific objectives of the nutritional intervention study. How can the authors establish a framework for food selection that caters to individuals with diverse nutritional requirements?

Q3. Line 79, the authors demonstrated that the data was obtained from available literature rather than a questionnaire or other form of self-acquisition. Is this paper a review or an article?

Q4. The references are found in the supplementary document, kindly make the necessary modifications. Supplementary documents typically contain supporting data only, while the analytical content can be presented in the main text. Additionally, it would be advisable to incorporate the data from the supplementary file into the main text.

Q5. Line 254, “whereas sugar sweetened beverages containing more nutrients”, please check the pertinent statements to identify which fruit juice or beverage is rich in nutrients.

Q6. The author's statement regarding the target population (adults or the elderly) should be consistently addressed throughout the entirety of the text, including in the title, rather than being limited to only a few paragraphs. Furthermore, the diet mentioned by the authors has been widely acknowledged for its health benefits across all age groups. How can the innovation be reflected? Why focus solely on adults or seniors? Where did the data in this paper come from, which several literatures or official data, it is suggested to supplement.

Author Response

Thank you for your constructive feedback and suggestions. We appreciate the time and effort you have put into reviewing our paper. Here’s how we plan to address your comments.

Q1. The excessive number of keywords necessitates further simplification.

Reply: We have revised the keywords to address your concern regarding their excessive number. The updated keywords are now more concise and focused: nutrient profiling; nutrient-rich foods index; convergent validity; hybrid nutrient density score; serving size; nutritional intake.

Q2. The abstract fails to accurately represent the target population and specific objectives of the nutritional intervention study. How can the authors establish a framework for food selection that caters to individuals with diverse nutritional requirements?

Reply: The Meiji NPS was not designed for individuals with specific health conditions.  Rather the Meiji NPS was intended for the Japanese population as a whole, with the only distinction made between adults and older adults. This approach is consistent with other existing NPS frameworks, which also focus on nutrient needs of populations as opposed to individuals. Those NP models are generally based on calories or weight as reference amounts.  We believe that NP models based on serving sizes, rather than per 100 grams, will provide better dietary guidance for the consumer. This approach also offers a more understandable and practical guideline for food manufacturers, helping to create a useful benchmark for potential product reformulation. We indicate this in Lines 270 to 273 and Lines 276 to 278.

Q3. Line 79, the authors demonstrated that the data was obtained from available literature rather than a questionnaire or other form of self-acquisition. Is this paper a review or an article?

Reply: Data on the size of food portions for each food and/or category were obtained from existing handbooks and documents.  For comparison purposes, data on serving sizes in the US can be obtained from the US Food and Drug Administration under the heading of Reference Amounts Customarily Consumed or RACC. Japan does not have government mandated portion sizes and data need to be obtained from multiple government and non-government sources. Otherwise, our paper is a research article with original data analyses.  As indicated in Line 59, support for the idea that consumers prefer NP models based on serving sizes came from published literature.

  1. Two papers are now cited. Kliemann, N.; Kraemer, M.V.S.; Scapin, T.; Rodrigues, V.M.; Fernandes, A.C.; Bernardo, G.L.; Uggioni, P.L.; Proença, R.P.C. Serving size and nutrition labelling: Implications for nutrition information and nutrition claims on packaged foods. Nutrients 2018, 10. DOI:10.3390/nu10070891.
  2. Van der Horst, K.; Bucher, T.; Duncanson, K.; Murawski, B.; Labbe, D. Consumer understanding, perception and interpretation of serving size information on food labels: A scoping review. Nutrients 2019, 11. DOI:10.3390/nu11092189.

Q4. The references are found in the supplementary document, kindly make the necessary modifications. Supplementary documents typically contain supporting data only, while the analytical content can be presented in the main text. Additionally, it would be advisable to incorporate the data from the supplementary file into the main text.

Reply: We have made the necessary modifications as per your suggestion in Line 200 to 205. The references that were previously found in the supplementary document have now been moved to the main text. We believe that the content presented in the supplementary document supports the main text comprehensively.

Q5. Line 254, “whereas sugar sweetened beverages containing more nutrients”, please check the pertinent statements to identify which fruit juice or beverage is rich in nutrients.

Reply: The sentence was modified to refer to sugar containing beverages. Generally, 100% juices contain free sugars but not added sugars. By contrast, juice based soft drinks and nectars (with 10% to 30% juice) typically contain added sugar. Thus 100% pineapple juice contains high levels of potassium and vitamin C and no added sugar, as also observed in other analyses cited here. By contrast, a peach based soft drink containing 30% juice does contain added sugar and can be lower in potassium and vitamin C. This is now clarified in our manuscript. We indicated this in Lines 264 to 267 in the revised manuscript.

Q6. The author's statement regarding the target population (adults or the elderly) should be consistently addressed throughout the entirety of the text, including in the title, rather than being limited to only a few paragraphs. Furthermore, the diet mentioned by the authors has been widely acknowledged for its health benefits across all age groups. How can the innovation be reflected? Why focus solely on adults or seniors? Where did the data in this paper come from, which several literatures or official data, it is suggested to supplement.

Reply: We appreciate your suggestions and have addressed them as follows:

  1. Consistency in Target Population: We acknowledge the importance of consistently addressing the target population throughout the text. We have revised the manuscript to ensure that the focus on adults and older adults is clearly stated in Line 95 in materials and method section.
  2. Innovation and Focus on Specific Age Groups: While the diet mentioned in our study may be beneficial across all age groups, our innovation lies in the development of the Meiji NPS for the two adult populations: adults and older adults (the elderly). We aim to utilize the Meiji NPS for product reformulation, providing direction and measuring the degree of improvement. In a separate study, higher values of the Meiji NPS for adults have shown a negative association with lifestyle-related disease indicators. Similarly, we are currently researching the association between the Meiji NPS for older adults and frailty-related indicators.
  3. Product Development: By using the Meiji NPS, we can develop products that address different health challenges across various life stages in Japan. Although we have currently developed the Meiji NPS focusing on adults and older adults, we plan to create a version tailored for children in the future.
  4. Data sources: Nutrient composition data were obtained from the Japanese Food Standard Composition Table 2020 (8th Edition), published by the Ministry of Education, Culture, Sports, Science, and Technology, Japan. There are no official values for serving size in Japan, thus we referred to the book “Ordinary serving values food composition tables,” which reported serving sizes as one or more edible quantities for a single food item.

We hope these revisions will address your concerns and improve the quality of our manuscript. Once again, thank you for your valuable input.

Reviewer 2 Report

Comments and Suggestions for Authors

The manuscript addresses significant issues. The text is well written. I have no significant objections. I do, however, have a few queries for the authors.

1. Has the distribution of the variables been verified? If both means and medians were presented, could the authors please clarify why this was done? In the event that this has not been done, it may be advisable to check the distributions of the variables and then select either the mean or the median.

2. Would it not be preferable to present the standard deviation value in accordance with the methodology described, as means ± standard deviations?

4. Please explain the rationale behind the choice of Speraman's R.

5. I am prompted to inquire about the rationale behind the high number of self-citations. I am aware that this paper represents a continuation of the authors' previous work. However, I believe that the number of self-citations may warrant further consideration.

Author Response

Thank you for your insightful comments and suggestion. We appreciate your positive feedback on our study. Here’s how we plan to address your comments.

  1. Has the distribution of the variables been verified? If both means and medians were presented, could the authors please clarify why this was done? In the event that this has not been done, it may be advisable to check the distributions of the variables and then select either the mean or the median.

Reply: We have chosen to use the median score for our discussion. The distributions of the serving sizes and the Meiji NPS scores are not normal distribution according to Shapiro–Wilk test and Kolmogorov–Smirnov test. Thus, we have removed means ± standard deviations in Tables 2 and 3, and Supplementary Table 1.

  1. Would it not be preferable to present the standard deviation value in accordance with the methodology described, as means ± standard deviations?

Reply: We have removed means ± standard deviations in Tables 2 and 3, and Supplementary Table 1. We have revised the manuscript accordingly to reflect this change.

  1. Please explain the rationale behind the choice of Spearman's R.

Reply: We have used the representative Meiji NPS score, which is based on the median. To align with this, we employed Spearman’s correlation, as it is appropriate for data that utilizes median values.

  1. I am prompted to inquire about the rationale behind the high number of self-citations. I am aware that this paper represents a continuation of the authors' previous work. However, I believe that the number of self-citations may warrant further consideration.

Reply: We would like to address the concern regarding the number of self-citations in our paper. In this manuscript, I have included two self-citations, both of which are essential for the context and understanding of our current work.

  1. The first self-citation is necessary to explain the classification of food groups to encourage within the Meiji NPS framework. This previous work provides the foundational categorization that is critical for the current study’s methodology and analysis.
  2. The second self-citation is required to elucidate the algorithm of the Meiji NPS. This prior research outlines the algorithmic approach that underpins the current study’s computational methods and results.

We believe that these self-citations are justified and integral to the continuity and comprehensibility of our research.

In addition, most of the papers by Dr. Drewnowski cited here date back to 2008 and 2009 and deal with the origins of nutrient profiling. For example, the paper on whether nutrient profiles ought to be based on 100 kcal, 100g, or serving sizes (essential for the present work) was first published in 2009. We have elected to cite the original literature rather than subsequent reviews.

We appreciate your understanding and consideration of this matter.

We hope these revisions will address your concerns and improve the quality of our manuscript. Once again, thank you for your valuable input.

Round 2

Reviewer 1 Report

Comments and Suggestions for Authors

No additional comments.